# 3D Printing of Pharmaceutical Application: Drug Screening and Drug Delivery

**DOI:** 10.3390/pharmaceutics13091373

**Published:** 2021-08-31

**Authors:** Ge Gao, Minjun Ahn, Won-Woo Cho, Byoung-Soo Kim, Dong-Woo Cho

**Affiliations:** 1Institute of Engineering Medicine, Beijing Institute of Technology, No. 5, South Street, Zhongguancun, Haidian District, Beijing 100081, China; gaoge@bit.edu.cn; 2Department of Mechanical Engineering, POSTECH, 77 Cheongam-ro, Nam-gu, Pohang 37673, Kyungbuk, Korea; mjahn@postech.ac.kr (M.A.); wwcho@postech.ac.kr (W.-W.C.); 3School of Biomedical Convergence Engineering, Pusan National University, 49 Busandaehak-ro, Mulgeum-eup, Yangsan 50612, Kyungbuk, Korea

**Keywords:** 3D printing, disease modeling, drug testing, drug delivery, drug screening, pharmaceutical application

## Abstract

Advances in three-dimensional (3D) printing techniques and the development of tailored biomaterials have facilitated the precise fabrication of biological components and complex 3D geometrics over the past few decades. Moreover, the notable growth of 3D printing has facilitated pharmaceutical applications, enabling the development of customized drug screening and drug delivery systems for individual patients, breaking away from conventional approaches that primarily rely on transgenic animal experiments and mass production. This review provides an extensive overview of 3D printing research applied to drug screening and drug delivery systems that represent pharmaceutical applications. We classify several elements required by each application for advanced pharmaceutical techniques and briefly describe state-of-the-art 3D printing technology consisting of cells, bioinks, and printing strategies that satisfy requirements. Furthermore, we discuss the limitations of traditional approaches by providing concrete examples of drug screening (organoid, organ-on-a-chip, and tissue/organ equivalent) and drug delivery systems (oral/vaginal/rectal and transdermal/surgical drug delivery), followed by the introduction of recent pharmaceutical investigations using 3D printing-based strategies to overcome these challenges.

## 1. Introduction

Since its introduction in the 1980s, three-dimensional (3D) printing has become a representative adjunct manufacturing technology. This technology was first developed for rapid prototyping and is widely applied in various industrial fields, including automotive, home appliances, space, and consumer goods [1]. In particular, advances in 3D printing techniques and the advent of printable/biocompatible materials have facilitated the fabrication of customized products in recent years [2]. Since the early 2000s, 3D bioprinting using biological materials such as cells and biomolecules has been successfully applied in tissue engineering to directly create living structures that reproduce the behavior of natural living systems [3]. The multiple advantages of 3D printing and bioprinting have become a major driving force for the rapid development of pharmaceutical applications, including drug screening and drug delivery systems (Figure 1A).

The development of new drugs is a high-cost and time-consuming endeavor, with high risks involved due to preclinical validation and clinical trials. Reportedly, an estimated 12–15 years are required for a single new drug candidate to undergo a series of evaluations before market availability (Figure 1B) [4]. For successful drug development, potential drugs are commonly identified and optimized through drug screening, followed by the selection of a candidate drug to progress to clinical trials. During the preclinical phase, drug candidates are screened for biological activity, toxicity, metabolism, pharmacological efficacy, and medicinal value using disease models for validating their efficacy and safety for further clinical trials. However, despite the prolonged evaluatory period, it has been reported that approximately 50% of failures can be attributed to unpredictable drug toxicity and inefficiency [5].

Over the past few decades, drug screening has conventionally depended on transgenic animals as disease models. However, the use of animal models raises serious ethical concerns, as well as confer inevitable limitations in precisely representing human tissue in the context of pathophysiology owing to genetic discrepancies [6]. Statistically, approximately half the drugs that have been proven safe in animal testing are found to be harmful to humans, while others remain nontoxic [7]. In an attempt to overcome the limitations of animal models, two-dimensional (2D)-based cell cultures have been employed as a screening platform for potential drug candidates. However, recent studies have revealed that cells cultured on a planar substrate differ not only in inherent phenotypes [8,9] but also in terms of cellular functions, such as migration [10], proliferation [11,12], and differentiation [13,14] when compared with cells cultured in 3D microenvironments. Accordingly, 3D printing improves the existing drug screening platforms by accurately depositing biomaterials containing patient-derived cells and can reproduce the natural environment of the diseased human body.

In addition to drug screening, the development of printable materials with various biodegradation profiles has facilitated the manufacture of drug delivery systems with individually controlled doses and customized medical devices that closely match the patient anatomical features [15]. A key objective of therapeutic administration is maintaining bloodstream drug levels between the maximum level that would cause toxicity and the minimum value that would be inefficacious in the body [16]. However, the plasma drug level rapidly increases after traditional dosing strategies, followed by a gradual decrease, necessitating successive dosing at regular intervals to maintain drug levels within an appropriate therapeutic window, thus resulting in patient inconvenience [17]. Moreover, most medical treatments are designed for the average patient as a one-size-fits-all approach, which might be successful in some but challenging to other patients. Furthermore, outcomes may differ among individuals in terms of drug efficacy or side effects. Traditional mass manufacture of drug delivery systems has failed to afford a cost-effective approach to address the diversity of therapeutic regimens caused by individual differences [18]. Therefore, current investigations assessing 3D printed drug delivery have focused on developing personalized drug delivery systems with controlled drug release profiles within the desired range, with minimal administration frequency.

This review provides a summary and brief classification of pharmaceutical applications applied to 3D printing technology. We first discuss the requirements of drug screening and drug delivery systems for advanced pharmaceuticals, which could be met by 3D printing. Furthermore, we describe the current methodological approaches of 3D printing applicable to the pharmaceutical field. Lastly, based on the strategy of 3D printing for more advanced drug screening and drug delivery systems, we introduce the recent pharmaceutical applications of 3D printing.

## 2. Requirements for Advanced Pharmaceutical Techniques

Pharmaceutical applications can be broadly divided into drug screening and delivery methods. The following sections classify and describe the elements required for advanced pharmaceutical technologies in each field of application.

### 2.1. Requirements for Drug Screening

Drug screening is employed to identify and optimize potential drugs prior to clinical trials to achieve an approved drug candidate. Drug screening allows the reduction of unnecessary time and costs by rapidly excluding unsuitable drug candidates from the initial stage of the validation. Moreover, the importance of screening for distinguishing drugs that can be effectively administered for therapeutic purposes is gradually gaining momentum. We discuss the essential elements of drug screening in the following sections.

#### 2.1.1. 3D Disease Modeling

Establishing a disease model that more accurately recapitulates pathophysiological behaviors and responses to a potential drug markedly improves the reliability of validation studies. Accordingly, the formulation of a 3D microenvironment reflecting actual tissues offers several advantages in drug screening, which can be achieved by encapsulating cells within the extracellular matrix (ECM). A 3D culture system enables physicochemical reactions that occur in native tissues to be executed by reproducing familiar in vivo conditions through cell–ECM interactions, as well as cell–cell interactions [19]. Li et al. revealed that fibroblasts cultured with a 3D matrix show three-fold adhesion sites when compared with those cultured using a 2D substrate, resulting in increased ECM secretion and cellular functions [20]. In addition, a 3D culture enables the observation of the spatiotemporal dynamics of nature, allowing the reproduction of metabolic exchanges and regulation of cell fate. For example, hypoxia is a crucial driver of cancer development and induces genomic stability and tumorigenesis [21]. In an attempt to generate central hypoxia, Yi et al. reproduced a radial oxygen gradient using a 3D gas permeable barrier with ECM containing brain cancer cells and endothelial cells, which is poorly reflected in a 2D culture platform [22]. Accumulating evidence has also demonstrated that spatiotemporal alteration of mechanical factors and adhesion peptides significantly impacts the guiding of eventual differentiation and disease progression [23,24,25].

Collectively, while 2D culture systems have limitations in representing complex tissue-level physiology, 3D culture conditions enable a more precise reproduction of tissue maturation by providing a native environment, enhancing cellular function, and promoting innate phenotypes. Accordingly, the development of 3D disease models that precisely simulate in vivo responses to potential drugs has gained considerable interest as a promising alternative to conventional disease models.

#### 2.1.2. High-Throughput Screening

Pharmaceutical development requires various considerations, including clear screening criteria for quantification, comparisons via counter screening, precise sequencing, drug metabolism processes, and response to false positives and negative outcomes [26]. These requirements further reinforce the need for numerous experimental samples. High-throughput screening (HTS), which is directly related to pharmacokinetics and toxicology, has been widely utilized in the pharmaceutical industry to effectively and efficiently achieve new drug development [27]. HTS analyzes the biochemical activity of hundreds of thousands of cell- or tissue-based samples, providing valuable information that reveals the effects of drug candidates on the disease of interest and the possibility of potential toxicity in non-diseased cells.

Conventional experimental samples are manually prepared by distributing small volumes of cell-containing suspensions into a 96-well or greater array using a multi-channel pipette [27]. However, this time-consuming process sometimes affords poor reproducibility, thereby requiring a high degree of automatization to discover ideal drug compounds among a multitude of experimental samples at high speed. Recently, HTS has been combined with liquid handling units, robotics, plate readers, and software programs for instrument control and data processing. Hou et al. generated pancreatic disease models on cell-repellent surfaces of 384- and 1536-well plates using an automated robotic system [28]. The disease model was developed using patient-derived pancreatic primary cancer cells, including cancerous fibroblasts, and evaluated the compatibility with HTS automation using over 3000 approved anticancer drugs. Accordingly, HTS saves time and costs by precluding poor candidate drug combinations in terms of efficacy, safety, and practicality in the early stage of drug development. Therefore, HTS has been widely employed for surveying cellular responses of potential drugs based on metabolism, pharmacokinetics, and toxicology to evaluate their effectiveness and potential adverse effects [29].

#### 2.1.3. Precision Medicine

Significant advances in precision engineering have confirmed the possibility of establishing customized tissue/organ models. In particular, precision medicine has served as a tool to help identify effective treatments based on patient-specific data, such as genetics, through ex vivo disease research [30]. Cells obtained from a patient can be employed to create an individualized model that more accurately represents specific patient diseases. This provides a more individualized assessment of pathological states than that afforded by generalized disease models using commercially available cell lines, as it considers even unique mutations that may vary among patients, such as genetic variation and cell-type mixtures [31]. Beager et al. illustrated that the constantly mutating nature of cancer cells renders predicting genotypes and determining appropriate treatments considerably challenging [32].

Genetic information closely related to multiple symptoms and disease pathologies contains vital information on disease behavior, enabling pharmacological studies of disease progression and drug treatment response [31,33]. For example, Kim et al. demonstrated that dermal fibroblasts obtained from a patient with type 2 diabetes notably differed in diabetes-related gene expression when compared with normal fibroblasts. The apparent diabetic features of the cells played a crucial role in developing an in vitro diseased skin model that could be employed for pharmacological investigations. Lee et al. confirmed patient-specific disease progression in vitro, based on differences in pathological characteristics and complex ecology of natural tumors using cells sourced from patients [22]. The authors assessed clinically observed outcomes in patient-specific resistance to chemoradiotherapy and medication using an in vitro disease model, suggesting appropriate drug combinations. Overall, the objective of precision medicine includes individualized diagnosis and therapy by utilizing the specific genetic, proteomic, and phenotypic features of a patient. In addition, precision medicine employed for drug screening improves monitoring of disease progression and response to potential drugs, conferring financial benefits.

### 2.2. Requirements for Drug Delivery System

Drug delivery systems are engineered devices or formulations that allow therapeutic agents to selectively reach the site of action. The development of an effective drug delivery system has the advantage of reducing potential off-target side effects, thereby maximizing efficacy. In the following sections, we discuss the requirements of drug delivery systems for pharmaceutical applications.

#### 2.2.1. Controlled Drug Release

Controlled drug release improves tolerability and reduces the incidence of side effects. Miller et al. demonstrated that the conversion of antiepileptic drugs from an immediate to a sustained-release formulation stabilized variabilities in absorption and improved tolerance to high doses exceeding 1200 mg per day [34]. The authors also revealed that sustained-release carbamazepine reduced the incidence of central nervous system side effects from 49% to 20%. Furthermore, premature drug dosing, rather than precise administration, could result in drug levels exceeding the maximum limit and inducing toxicity; however, a missed dose can reduce treatment efficacy. Controlled drug release reduces fluctuations in the released concentration of therapeutic agents throughout the administration period, increasing bioavailability by conferring highly predictable and reproducible release kinetics [35]. Chai et al. printed a hollow hydroxypropyl cellulose tablet loaded with domperidone, an insoluble weak base, to improve bioavailability [36]. This 3D printing formulation demonstrating floating sustained-release profiles was established by analyzing the release kinetics according to the proportion of materials used. Accordingly, when administered to rabbits, the 3D printed formulation was retained for more than 8 h, revealing an apparent difference in bioavailability of more than two-fold when compared to that of commercial tablets without employing 3D printing techniques.

Administered drugs are exposed to diverse environments depending on their location, method, and metabolic pathway. Accordingly, 3D printing strategies for controlled drug release involve the regulation of drug release profiles by printing a drug carrier composed of a material that degrades in response to specific stimuli to protect the drug until it is delivered to the region of interest. For example, Okwuosa et al. and Goyanes et al. developed 3D printed oral drugs for intestinal illness by using enteric polymers such as methacrylic acid [37] and hypromellose succinate [38]. These 3D printed tablets using enteric polymers do not degrade under acidic conditions in the stomach and initiate drug release in the basic environment, which can effectively enable drug delivery to the small intestine. This approach can be used to significantly reduce the frequency of prescriptions, resulting in improved comfort and compliance among patients who require frequent and repeated drug administration to inaccessible areas such as the joints, heart, or eyes.

#### 2.2.2. Personalization

In order to overcome the limitations of conventional mass production, targeted drug delivery is attracting attention as a personalized therapy to tailor disease prevention and treatment by accounting for differences in each patient or lesion site. Recently, advances in biomedical technology have enabled the on-demand manufacture of individual geometrics through the rapid optimization of multiple parameters. Robinson et al. introduced a fabrication approach for the rapid prototyping of a customized implant to treat bronchomalacia in newborns [39]. The authors also established highly variable geometries with superior quality based on computed tomography (CT) scan images of the patient. Therefore, advances in flexible technology to fabricate complex geometries are expected to provide an avenue for patient-specific treatment by employing anatomical information of the patient to produce formulations containing specific drugs [40].

Although prescribing each active pharmaceutical ingredient separately to treat a combination of diseases remains a commonly employed strategy, this inconvenient approach might lead to medication errors and serious patient compliance problems [41,42]. To address this issue, combining several therapeutic agents into a single drug delivery system with an individually optimized release profile and dosage has been presented as an attractive alternative [43]. Reportedly, a combination of hypotensive agents at low doses, including statins, aspirin, and folic acid, reduced cardiovascular disease by more than 80% [44]. Based on these results, a commercial single drug carrier with a drug combination essential for treating cardiovascular disease was developed under the name Polycap™ [44,45]. Similarly, Khaled et al. developed a five-component drug delivery system physically separated by cellulose acetate, designed as a permeable carrier using a mixture of excipients [46]. The cellulose acetate was first extruded to physically separate the drugs that were printed through the respective printing cartridges. The immediate release compartments of aspirin and hydrochlorothiazide were subsequently printed on the upper part of the drug system to cover the sustained-release compartments. The developed system allowed each loaded drug to be manipulated by a different release profile, achieving drug delivery for various purposes via a single delivery vehicle.

## 3. Advent of 3D Printing for Pharmaceutical Application

3D printing is an additive manufacturing process for creating three-dimensional objects from a digital file. 3D bioprinting has been recognized as a promising technology for creating tissue-based platforms with high reproducibility and versatility by accurately positioning biomaterials together with cells and biomolecules. These features of 3D bioprinting are directly associated with requirements of drug screening and drug delivery systems, enabling the design of more advanced pharmaceutical applications. To meticulously design a physiologically functioning model/device, researchers should consider various aspects of 3D bioprinting, such as suitable biomaterials, cell sources, and printing strategies.

### 3.1. Bioink

The appropriate choice of printable biomaterials, commonly referred to as bioinks, is essential for building tissue architectures with desired biophysical and biochemical properties. Biomaterials currently employed as bioinks are predominantly natural or synthetic-based polymers, which should possess features of biocompatibility and printability. In selecting an appropriate bioink, the major features of each bioink should be considered.

Naturally derived biomaterials offer greater similarity to biophysical and biochemical constituents in native tissues, thereby closely recapitulating biological responses when compared with synthetic biomaterials. Numerous natural polymers isolated from animal or human tissues have been developed as bioinks and have revealed superior cell affinity to promote cellular functions such as migration, proliferation, and differentiation [47]. However, most natural bioinks, including collagen [48], gelatin [49], alginate [50], and hyaluronic acid [51] possess only a single protein component of the ECM and remain limited in representing intrinsic biophysical and biochemical elements such as growth factors, glycosaminoglycans, laminin, fibronectin, and elastin [52]. From this perspective, recent studies have highlighted the significance of decellularized ECM (dECM) as a promising bioink, allowing a niche microenvironment with synergistic effects on encapsulated cells [53,54]. Based on the differential proteomic analysis, Han et al. confirmed that a variety of inherent matrisome protein constituents were retained in each dECM bioink, playing a crucial role in inducing tissue-specific cellular behavior [55]. Although natural polymer-derived bioinks can provide better cell affinity, their weak mechanical properties hinder the construction of cellular architectures. To compensate for the innate limitations of naturally derived bioinks, several researchers have utilized synthetic polymers that can improve or modulate a wide range of features, including mechanical properties, degradation profiles, crosslinking mechanisms, and chemical compositions [56]. Huston et al., for instance, developed a photocrosslinkable composite hydrogel by incorporating poly(ethylene glycol) (PEG) and methacrylated gelatin (GelMA) (i.e., denatured collagen), and demonstrated that its biological and mechanical properties could be adjustable by altering the concentration of each component [57]. As an alternative approach, Pati et al. showed that 3D bioprinting using multiple bioinks can compensate for the weak mechanical properties of naturally derived polymers by first depositing synthetic polymers that serve as a supportive framework [58].

Selecting a proper bioink depends on various factors, such as the characteristics of the target tissue, printing strategy, and biological process. To effectively build a drug screening platform and drug delivery device, it is necessary to continue developing and optimizing physically and chemically tunable biomaterials.

### 3.2. Cell Source

Although most drug delivery devices are acellular, appropriate cell selection is a paramount factor in designing a drug screening platform to represent the physiological state and pathological process of the tissue of interest more precisely. Three types of cell sources (primary cells, cell lines, and stem cells) are commonly used to create a 3D bioprinted cellular model. To narrow the gap in biological responses due to genetic discrepancies between animals and humans, most in vitro platforms for disease modeling prefer utilizing cells derived from human tissue [59,60,61]. Among various cell types, primary cells directly isolated from the tissue have the advantage of reproducing tissue functions at specific points or stages [62]. However, several challenges concerning the limited lifetime and donor-to-donor variations need to be resolved [63]. Cells that can continuously propagate over a prolonged period are called cell lines and have homogeneous phenotypic and genotypic features. These immortalized cell lines can proliferate indefinitely through genetic mutations or artificial modifications. Compared with other cell types, cell lines can be purchased at a low price and allow convenient handling. Accordingly, cell lines can be ideal for testing cells to establish a new tissue fabrication platform. However, cell lines are less preferred as a biologically relevant option because they lose the inherent characteristics of the original tissue. Hence, cell lines are not preferably used for the development of personalized artificial tissues. The third type of cell commonly used in 3D bioprinting is stem cells, including mesenchymal stem cells, embryonic stem cells, and induced pluripotent stem cells (iPSCs). Stem cells characterized by self-renewal and differentiation potency are gradually attracting attention due to the unlimited potential for tissue regenerative medicine and in vitro human disease modeling. In particular, the ability to reprogram donor-specific properties enables an improved understanding of phenotypic variability and disease mechanisms, possibly providing an accurate solution focused on a specific patient [64]. Stem cells with different lineages and potencies have been widely utilized in 3D bioprinting for human disease modeling. For example, Dai et al. created a brain tumor model using glioma stem cells, known to be the primary cell type closely associated with high-grade gliomas [65]. The researchers verified that stem cells composing the brain tumor model maintained their intrinsic characteristics while affording differentiation potential during the entire in vitro culture period. However, the ability of stem cells to precisely regulate the differentiation pathways into desired lineages and the immature cell phenotype genetically similar to fetal cells needs to be addressed [62].

Regardless of the nature of the cell source, there is a high possibility of variations between batches, attributed to various causes such as technical skills of users, differences in culture conditions [66], and inconsistent cell differentiation potency [67]. Therefore, it is essential to characterize cells through multiple assays and approaches to reduce inconsistencies and variations between experiments [68,69].

### 3.3. Printing Strategy

The 3D printing strategy for biomaterial deposition includes inkjet-based, extrusion-based, and laser-assisted printing. These approaches possess different features, including the printing mechanism, resolution, speed, and applicable biomaterials. Herein, we provide a brief overview of the respective printing strategies.

#### 3.3.1. Inkjet-Based 3D Printing

Inkjet-based 3D printing is a non-contact approach that allows controlled volumes of liquid bioink to be precisely dispensed onto a planar substrate (Figure 2A). Inkjet printers employ thermal, piezoelectric, or electromagnetic methods to apply bioinks drop-wise from a nozzle. Thermal inkjet printers electrically heat the printing head up to 300 °C, inflating an air bubble to eject the droplet. It has been confirmed that markedly high temperatures do not have harmful effects on biological molecules or cells in the bioink because localized and momentary (~2 µs) heating increases the overall temperature to only 4–10 °C [70]. Piezoelectric inkjet printing creates droplets at regular intervals by generating pressure with an acoustic wave inside the printing head, whereas the electromagnetic approach uses electromagnetic forces such as Lorentz or permanent magnetic configurations to position the droplets.

Inkjet printing can regulate the amount of bioink ejected with a high resolution of up to 20 μm [71,72]. Using a commercial inkjet printer, Roth et al. achieved highly precise cellular patterns in various shapes by depositing biologically active proteins, which modulate cell attachment [73]. In addition, Christensen et al. demonstrated the feasibility of inkjet bioprinting to build a cell structure similar to that of blood vessels with horizontal and vertical branches [74]. Inkjet bioprinters enable the fabrication of multiple tissues within a limited area. For example, Xu et al. successfully produced a complex heterogeneous construct composed of three different cell types [75]. Recent studies have demonstrated the potential of the inkjet printing technique in HTS for drug discovery, as it offers miniaturization, repeatability, low costs, high resolution, and generation of minimal waste and contamination [76]. Furthermore, the inkjet-based approach is widely applied for creating a plethora of microarrays for genomic and pharmacological profiling of a biochemical pathway of interest [76]. Hughes et al. developed a microarray system for gene expression profiling using oligonucleotides synthesized using an inkjet bioprinting-based approach [77].

Although inkjet bioprinting has gained momentum in recent decades owing to its versatile potential, there remain limitations that need to be solved in the near future. As only bioinks with low viscosity (~0.1 Pa∙s) can be applied in inkjet printers, the narrow range of printable bioinks has often been challenging in inkjet-based bioprinting [78]. In addition, its drop-by-drop biomaterial deposition might prolong the printing time, resulting in the increased likelihood of nozzle clogging [79].

#### 3.3.2. Extrusion-Based 3D Printing

Extrusion-based 3D printing continuously deposits a stream of bioink on a substrate driven by pneumatic [72,80,81] or mechanical [82,83,84] dispensing systems (Figure 2B). The pneumatic system is the most commonly utilized extrusion method owing to its low cost and ease of use. It uses compressed air to push the bioink, while the mechanical method employs a piston or screw to force the materials to be extruded through a nozzle. However, a mechanically driven approach can precisely regulate the amount of extruded material when compared with the pneumatic system, causing a delay time due to the compressed air volume. Unlike inkjet-based 3D printing that requires low-viscosity bioink, both extrusion mechanisms can employ a wide range of high-viscosity bioinks (0.03–6 10^4^ Pa∙s), including dECM, gelatin, alginate solution, and thermally or chemically molten synthetic polymers [85,86,87,88]. Undoubtedly, the use of bioink that has high mechanical strength affords the possibility of fabricating a final tissue architecture with more robust mechanical properties.

Extrusion-based 3D printing allows the selective printing of multiple types of bioinks in a predefined location [89,90], which also contributes to simplifying the fabrication process by simultaneously printing cellular components and structural support [91]. More recently, extrusion-based 3D printing was shown to achieve fiber-shaped tubular constructs using a coaxial nozzle [92,93], which simultaneously extrudes alginate-laden bioink and Ca^2+^-laden fugitive bioink via the shell and core of a coaxial nozzle, respectively.

It is essential to consider the cell damage inflicted from shear stress during nozzle passage in extrusion-based 3D printing [94]. Accordingly, to protect cells from stress during passage through the nozzle, bioinks should exhibit shear-thinning behavior, reducing viscosity under shear strain. This property decreases the stress inflicted on cells and maintains the printed shape [53]. Furthermore, although low pressure and a large nozzle size probably maintain high cell viability, the trade-off is a limited resolution and printing speed. Hence, researchers should establish diverse printing conditions that barely influence cell viability and additional functionalities before producing the final tissue product.

#### 3.3.3. Light-Assisted 3D Printing

Light-assisted 3D printing is divided into laser-induced forward transfer (LIFT) and stereolithography (STL) techniques. First, LIFT consists of a pulsed laser, an energy-absorbing support, and a substrate where the bioinks are deposited (Figure 2C). The energy-absorbing support was coated with a cell-laden hydrogel at the lower surface. This technology provides a high printing resolution by controlling the droplet transfer to the receiving substrate. When the laser is focused on the energy-absorbing layer, the energy absorption generates a vapor pocket or mechanical wave that forces the droplet to be separated from the support. The resolution of light-assisted printing is generally known to range between 20 and 30 μm and is associated with several parameters such as energy per surface, viscosity of the bioink, surface tension, and the thickness of the energy-absorbing support [95]. Unlike inkjet- or extrusion-based approaches, as this strategy does not require any printing nozzles, it can avoid the issue of nozzle clogging with biological materials. Despite its superior capability in affording high cell viability and resolution, this system is limited due to the operational costs of laser-assisted printers and preparation of printing components such as the energy-absorbing layer and collecting substrate [96].

STL, one of the oldest 3D printing technologies, enables the creation of complex geometry in a layer-by-layer fashion by selectively solidifying liquid photopolymers with ultraviolet (UV), infrared, or visible light (Figure 2D) [97]. This technology projects a 2D slice containing cross-sectional information of a 3D model from biomedical information, such as magnetic resonance imaging (MRI) or CT, onto a photopolymer reservoir, thereby allowing the fabrication of a volumetric and arbitrary structure at a rapid speed when compared with a biomaterial deposition-based approach through a printing nozzle. The focal size of the light source determines its printing resolution, which is usually at the microscale level [98]. High shape fidelity with rapid fabrication speed has enabled researchers to produce highly elaborate structures useful for biomedical applications. More recently, the two-photon polymerization-based approach, a direct-writing technique, has been introduced as a promising new technology (Figure 2E). When the laser is focused at a single point in the photosensitive monomer, it begins to polymerize by simultaneously absorbing two photons. The high feature resolution due to the unique polymerization mechanism has allowed the fabrication of biomedical devices at micro/nanoscale sizes [99]. Gittard et al. successfully developed microneedle array templates for the transdermal delivery of protein-based pharmacological agents, and the dimensions of each microneedle were hundreds of micrometers [100,101]. Initially, as photocurable materials in STL were unsuitable for use with living cells, they were utilized to build patient-specific models of interest as surgical guides to reduce time and potential risks in the operating theater [102,103]. However, biocompatible/photosensitive materials such as collagen type I [104], laminin [105], streptavidin [106], and PEG-based hydrogels [107] have been successfully developed in recent years. Many researchers have created implantable and anatomically patient-specific devices for the skin, heart valves [108], aortas [109], and nasal implants [110] using the STL-based approach.

Light-assisted 3D printing has a significant advantage in constructing complex structures rapidly, but it remains challenging to build a heterogeneous construct consisting of multiple materials. Thus, the selection of an appropriate printing strategy should be considered based on the characteristics of each printing technology.

## 4. Drug Screening

To evaluate the pharmaceutical safety and therapeutic efficacy of drug candidates, in vitro analytical platforms should precisely recapitulate the anatomical characteristics and critical functions of target tissues/organs. Owing to the advantage of flexibly positioning cells, biomolecules, and biomaterials, 3D bioprinting techniques enable the establishment of advanced 3D cell culture devices that can be typically divided into three typical subjects: organoid, organ-on-a-chip (OOC), and tissue/organ equivalent (Figure 3). This section summarizes the strategies for grafting 3D bioprinting techniques to engineer these advanced analytic systems, review their applications for drug screening, and discuss their advantages and limitations.

### 4.1. 3D Bioprinted Organoid

#### 4.1.1. Organoid

Organoids, defined as miniature organs, are derived from the 3D culture of mammalian tissue-resident stem/progenitor cells or embryonic stem cells in the presence of necessary physiological cues and matrices. Organotypic constructs rely on cell proliferation, self-organization, and differentiation into specific cell lineages, and multiple types of cells resembling both the microarchitecture and functional characteristics of the target organ can be generated. Bissell et al. reported the first organoid structure in 1982, which was derived from both normal and tumor murine mammary glands by cell self-assembly in the presence of laminin and collagen IV [111]. Subsequently, pioneering studies by Clevers et al. and Sasai et al. successfully developed intestinal and cortical organoids using adult stem cells and iPSCs, respectively [112,113]. To date, a variety of human organoids (e.g., brain, retina, lung, liver, kidney, pancreas, stomach, and intestine) have been used to investigate infectious diseases, genetic disorders, and cancer [114].

Although the traditional 3D culture of stem cells has been widely used to produce organoids due to its simplicity, this methodology has several critical limitations limiting the application of organoids. In particular, the random configuration of traditional 3D culture lacks precise control of organoids in terms of dimension and complexity. Challenges in reproducibility significantly limit the standardization and industrial production of organoids. In addition, it is difficult to incorporate complex and dynamic microenvironments (e.g., dynamic flow and mechanical stimulus) found in organ morphogenesis as instructive cues during organoid assembly. Therefore, to increase the complexity of organoids, challenges concerning the reproduction of 3D cellular structure and recruitment of perfusion networks need to be addressed. Owing to the ability to precisely position cells and biomaterials in a specific environment to create living constructs that closely imitate natural tissue and organs, 3D bioprinting techniques demonstrate unparalleled potential for overcoming the current challenges of organoid fabrication.

#### 4.1.2. Strategies for 3D Bioprinting of Organoids

Although 3D bioprinting of organoids is still in its infancy, undifferentiated pluripotent stem cells or differentiated stem cells have been biofabricated using extrusion and light-based bioprinting techniques to generate organoids in vitro. Gu et al. extruded human iPSCs within a hybrid bioink composed of alginate, carboxymethyl-chitosan, and agarose, differentiated in situ to self-assemble 3D embryoid bodies expressing three germ markers [115]. In a recent report, Kim et al. directly bioprinted a high density of human metastatic melanoma cells (1 × 108 cells/mL) within dermis-specific dECM bioink into a supporting bath in a point-by-point manner (Figure 4A) [116]. Owing to the flexibility of 3D bioprinting, cell-laden spheroids with varied dimensions and pre-designed patterns could be readily achieved, which further generated organoids of melanoma. This bioprinting strategy provides an expedient method to spatially orchestrate organotypic structures, which helps improve the complexity of organoids and produce HTS platforms. In addition to extrusion-based 3D bioprinting, Yu et al. developed a modified dECM bioink that can be photopolymerized based on UV irradiation and combined STL technique to build microscale biomimetic tissue constructs with human iPSC-derived cardiomyocytes and hepatocytes, which exhibited high viability and maturity during determined culture periods (Figure 4B) [117]. Several recent studies have suggested that the process of 3D bioprinting might not have a detrimental effect on the pluripotency and differentiation lineage of stem cells [118,119].

In addition to the biofabrication technique, accumulating evidence has demonstrated that the composition of the applied bioink plays a vital role in ensuring long-term cell viability while offering stem cell niches for guiding cell differentiation. For example, Nguyen et al. compared the performances of human iPSCs co-cultured with chondrocytes in bioinks consisting of nanofibrillated cellulose (NFC) with alginate and hyaluronic acid [120]. Compared with the well-maintained pluripotency and cartilaginous tissue in NFC/alginate bioink for five weeks, regressed proliferation and pluripotency of the cells in NFC/ hyaluronic acid were observed. Hence, the selection of bioink materials is critical for the 3D bioprinting of organoids. To date, researchers have utilized various combinations of ECM-based bioinks to investigate their effects on guiding stem cell fates, including hyaluronic acid, laminin, gelatin-methacrylate, and tissue-specific dECM. However, as our knowledge regarding stem cell self-organization for organ morphogenesis remains limited, substantial work is needed to establish a standardized bioink pool for grafting 3D bioprinting to organoid construction.

Except for the living cell-embedded bioink, cell aggregates or even organoids can be used as biomaterials for 3D bioprinting. For instance, a recent study developed a new bioprinting approach to transfer pre-cultured cellular spheroids into self-healing support hydrogels at high resolution, which enabled their patterning and fusion into high-cell density microtissues of a predetermined spatial organization (Figure 4C) [121]. In another study, tissue-specifically differentiated colon organoids were embedded within the GelMA bioink and 3D extruded into 96-well culture plates, maintaining viability and proliferative capacity post-print [122]. Despite successful attempts, the direct bioprinting of organoids faces additional challenges when compared with the bioprinting of cell-laden bioink. One reason is that organoids exist in the form of cell aggregates. Hence, unlike cells dispersed in the bioink system, it is difficult to dispense an organoid through a conventional bioprinter nozzle. In addition, it is necessary to elucidate the adverse effects of mechanical force, impact, heat, chemical components, and irradiation during the bioprinting process on the maintenance of organoid viability and function. Therefore, relevant fundamental investigations should be conducted before applying organoids as bioprintable elements.

#### 4.1.3. Pharmaceutic Applications of 3D Bioprinted Organoids

By incorporating the 3D bioprinting technique, various organoids have been developed for regenerative medicine, disease modeling, and drug screening. The liver and kidney are critical organs for detoxification and systematic metabolism, attracting considerable interest in organoid construction to assess drug toxicity and metabolic processes. Yang et al. utilized extruded HepaRG cells encapsulated in a bioink composed of alginate and gelatin to construct 3D bioprinted hepatorganoids in vitro and demonstrated the detoxification function (Figure 4D) [123]. Moreover, upon implantation into a mouse model of liver injury, the bioprinted hepatorganoids further matured and functionalized, improving the survival rate of mice by affording human-specific drug metabolism.

Notably, 3D bioprinted organoids could address the limitations of 2D cell culture and animal models for in vitro disease modeling, given their ability to reproduce both cell–cell and cell–ECM interactions. Various bacteria/virus-induced intestinal organoid models have been developed. During the coronavirus disease (COVID-19) pandemic, human organoids have been employed to elucidate the pathology and develop effective treatments. Pioneering works evaluated how severe acute respiratory syndrome coronavirus 2 (SARS-CoV-2) infects and damages human organs by using organoids of the human eye [124], airway [125], intestine [126], kidney [127], and brain [128]. However, as organoids only recapitulate the representative similarity of human organs, the lack of key in vivo features such as functional immune system, organ–organ interaction, and homeostasis narrows its current application, merely affording a complementary approach to animal disease models.

Organoids created from patient-derived stem cells possess unique merits for understanding hereditary diseases and investigating optimal therapies for individual patients. Patient-derived liver organoids have enabled the modeling and study of disease pathology of genetic liver diseases, including α1-antitrypsin deficiency and Alagille syndrome [129]. Undoubtedly, patient-specific organoids would provide the best models for dissecting complex disease mechanisms in vitro. Equipped with 3D bioprinting, a promising technique for complex organoid construction, extensive applications of organoids in biomedical and clinical fields are foreseeable.

For drug discovery, equipped with a 3D bioprinting technique, it is possible to construct massive and reproducible organoids. This advantage allows modulating the microenvironment with high freedom. Reid et al. demonstrated the precise generation of tumoroid arrays and positioned cancer cells within chimeric structures using a 3D bioprinter, facilitating the study of tumorigenesis and microenvironmental redirection of breast cancer cells (Figure 4E) [130]. In addition, the high reproducibility of organoids is conducive to constructing multiple samples at industrial levels for HTS. Lawlor et al. replaced the manual production of kidney organoids with automatic bioprinting of iPSCs into multiple-well plates, which improved throughput, quality control, scale, and structure [131]. This achievement would leverage the strength of 3D bioprinting technology for in vitro and in vivo applications of organoids.

### 4.2. 3D Bioprinted Tissue/Organ Equivalent

#### 4.2.1. Tissue/Organ Equivalent

In contrast to organoids generated by morphogenesis following population and differentiation of self-renewal stem cell colonies, the tissue/organ in vitro equivalent was directly constructed to mimic the anatomical and physiological characteristics of their natural counterparts. The tissue/organ equivalent has emerged as a promising tool for drug toxicity assays and disease modeling, given the advantages of reproducing complex key human physiological aspects. Although conventional tissue engineering techniques can combine scaffolds and cells to generate living tissues for medical applications, traditional methods are limited to constructing structures with precise architectural features and spatial localization of multiple types of cells, hindering the generation of tissue/organ equivalents. As an additive fabrication technique, 3D bioprinting can produce a construct with complex structures and heterogeneous compositions, which emulates native tissues and organs. These constructs are valuable for drug discovery, precise drug application, and personalized drug screening.

#### 4.2.2. Strategies for 3D Bioprinting of Tissue/Organ Equivalent

Although organotypic bioprinting could pave the road ahead for fabricating biomimetic tissues/organs in vitro, several critical challenges should be overcome before achieving applicable tissue/organ equivalents. First, the tissue/organ equivalent should replicate the anatomical and structural complexity of natural targets, such as spatial localization of different tissues, as well as their interfaces, vital for executing regular tissue/organ functions. For example, human skin is composed of dermal and epidermal layers. The signal crosstalk between resident fibroblasts and keratinocytes is critical for epidermal differentiation and maturation [132]. Similarly, the interactions between the intima and media tissues in human blood vessels play an important role in regulating vascular functions (e.g., vascular tone and cellular phenotypes) [133]. Hence, compartmentalized organization and tightly correlated interfaces are pivotal considerations when constructing layered tissues in vitro. Simulating complex architectures and distributions requires the precise deposition of bioinks. However, most natural ECM-based bioinks, which are widely employed for 3D bioprinting, exhibit unadoptable printability owing to their low viscosity and slow gelation speed. Hence, to construct tissue and organ mimics using these bioinks, several advances in 3D bioprinting strategies have been achieved, including accelerated gelation (e.g., crosslinker aerosol spray [134], coaxial extrusion [93], heating assisted printing [135]) and physical support (e.g., multiple-head collaborative biofabrication [86] and suspended 3D bioprinting [136]).

Secondly, the vasculature, i.e., the circulatory system in the human body, is closely related during a large group of diseases (e.g., coronary heart disease and cancer) and is the site of action of several drug candidates (e.g., anti-viral, anti-tumor, anti-inflammation, and anti-angiogenesis). In addition, a hypoxic microenvironment may develop in the fabricated tissue analogs owing to the limited diffusion of nutrients. Reportedly, cells located distant from blood vessels (>200 μm) showed significantly decreased viability [137]. Therefore, vascularization is typically indispensable for modeling functionally viable and long-term stable tissue/organ equivalents. 3D bioprinting can localize growth-factor-secreting cells and biomolecules within a construct to program gradient signals that can induce spontaneous angiogenesis and vasculogenesis, generating perfusable vascular networks [138]. Moreover, the fabrication of perfusable channels using coaxial printing, template printing, and direct STL is a widely used 3D bioprinting strategy for engineering vascular networks in tissue/organ equivalents [139].

Third, the human body is a complex environment entailing various physiological stimulations, such as chemical, physical, mechanical, and electrical signals. In order to better recapitulate the function and model disorders of tissues and organs, these microenvironmental elements should be considered. For instance, a regular laminal bloodstream maintains the quiescent phenotype and stable function of endothelial cells in blood vessels, while turbulent flow induces endothelial dysfunction that leads to the initiation of atherosclerosis [140]. Gao et al. applied an in-bath coaxial printing technique to build a geometry-tunable artery equivalent and understand the role of hemodynamics in regulating pathophysiology [93]. By altering the shape design of the equivalent, laminar and turbulent flows were generated in regular and stenosis/torturous models, resulting in varied cellular responses and pathological events, respectively. As another example, human skeletal muscle is a highly innervated tissue; without electrical stimulation provided by connected motor neurons, the denervated skeletal muscle could become atrophic, most often leading to patient death [141]. Therefore, neuron integration is critical for engineering skeletal muscle equivalents. Kim et al. coprinted human muscle progenitor cells and human neural stem cells to construct a neural-integrated skeletal muscle [142]. The involvement of neural neurons in the skeletal muscle construct promotes myofiber generation, long-term survival, and neuromuscular junction formation in vitro. Although 3D bioprinting has demonstrated its versatility for fabricating advanced tissue/organ equivalents with improved physiological relevance, the construction of millimeter/centimeter-scale structures usually requires a considerable number of cells. In addition, to build a complex tissue analog, the bioprinting process commonly involves multiple preparative and implementation procedures. These drawbacks limit the medical and clinical applications of 3D bioprinted constructs at an industrial level.

#### 4.2.3. Pharmaceutic Applications of 3D Bioprinted Tissue/Organ Equivalents

To date, these innovative strategies have encouraged researchers to successfully construct a variety of tissue/organ equivalents that resemble the complex anatomical and pathological features of the simulated targets, including the heart [143], kidney tubules [144], blood vessels [93], skin [145], cornea [146], airway [147], pancreas [148], skeletal muscle [136], and gastrointestinal tract [149,150]. The developed tissue/organ physiological and disease models have been applied for drug discovery/screening and pathological understanding. The representative progress and achievements are summarized and categorized in Table 1.

**Table 1 pharmaceutics-13-01373-t001:** Representative pharmaceutic applications of 3D bioprinted tissue/organ equivalents.

Physiological Tissue/Organ Equivalents				
		**Printing Strategy**	**Achievement**	**Reference**
Foreign barrier	Skin	Integrated extrusion-inkjet 3D bioprinting	Full-thickness skin model with improved physiological relevance; perfusable vascularized skin equivalent composed of epidermis, dermis, and hypodermis.	[151]
	Airway	Indirect STL-based 3D bioprinting	Indirectly printed and reinforced with silicone rubber for creating a native mimetic tracheal framework; stratified mucosal layer formation by transferring stem cell sheets onto the luminal surface.	[147]
	Cornea	Extrusion-based 3D bioprinting	Arrangement of anatomically relevant corneal fibrillar structures controlled by printing nozzle size and shear stress.	[146]
Circulation	Vessel	Extrusion-based coaxial 3D bioprinting	Fabrication of freestanding, perfusable, and functional in vitro vascular model.	[93]
	Heart	Extrusion-based suspended 3D bioprinting	Thick, vascularized, and perfusable cardiac equivalent matching the immunological, cellular, biochemical, and anatomical characteristics.	[143]
	Renal	Extrusion-based tri-coaxial 3D bioprinting	Microfluidic tubes that recapitulates tubular/vascular renal parenchyma composed of renal tubular epithelial and endothelial cells.	[144]
Digestion	Intestine	STL-based 3D bioprinting	Engineering of intestinal structure with a crypt/villus architecture and tissue polarity by combining a photopolymerizable hydrogel with a high-resolution STL technique.	[149]
	Pancreas	Extrusion-based 3D bioprinting	Engineered pancreatic equivalent consisting of insulin-producing cells encapsulated in pancreatic tissue-specific bioink.	[148]
**Diseased tissue/organ equivalents**				
		**Printing strategy**	**Achievement**	**Reference**
Diabetes		Integrated extrusion-inkjet 3D bioprinting	3D diseased skin tissue with pathophysiological features of type 2 diabetes in vitro; crosstalk between diabetic fibroblasts and epidermal keratinocytes to promote diseased epithelial morphogenesis.	[145]
Atherosclerosis		Extrusion-based in-bath coaxial 3D bioprinting	Direct fabrication of three-layered arterial-mimetic tubes with stable mechanical properties; recapitulation of various stimulation inducing endothelial dysfunction by stenotic and turbulent flows.	[140]
Cancer		Extrusion-based in-bath 3D bioprinting	3D tumor mimetic construction consisting of a metastatic cancer unit and a perfusable vascular system by a tissue-level fabrication printing platform; metastasis-associated changes by precisely controlling distal regions.	[116]

### 4.3. 3D Bioprinted Organ-On-A-Chip

#### 4.3.1. Organ-On-A-Chip (OOC)

OOC is broadly defined as a microfluidic cell culture device that models the functional units of human organs. Microchannels allow not only the control of nutrient supply but also the flexible manipulation of biomechanical microenvironments (e.g., stress force, fluid flow, cyclic motion) and biochemical features (e.g., chemotaxis and oxygen gradients). Moreover, the microfluidic technique can achieve the effective spatiotemporal delivery of pharmaceuticals/biomolecules and in situ monitoring of cellular responses, facilitating the implementation of versatile and real-time drug screening. Furthermore, as a goal of OOC, combining multiple chips representing different organs can reconstitute a body-on-a-chip to mimic organ–organ interactions that regulate the drug delivery process, such as drug metabolism and absorption at the systemic level. Given these merits over 2D cell culture models, OOC is considered a potential analytic tool for pathological studies and drug screening.

The construction of a viable OOC relies on understanding the anatomy of the target organ and simplifying it to fundamental elements that are essential for physiological functions. A representative example is a lung-on-a-chip device, which includes two adjacent microchannels separated by a thin and flexible porous membrane [152]. While the implemented vacuum triggers microchannel motion to simulate the cyclic breathing strain, the membrane creates an air–liquid interface that facilitates the separate coculture of epithelial cells and endothelial cells, mimicking the alveolar-capillary lung interface. As it can successfully recapitulate organ functions and reflect pathological changes toward given stimulations, lung-on-a-chip has been applied to analyze drug efficacy, dosage responses, and toxicity. Similarly, various tissues and organs such as the liver, kidney, brain, heart, blood vessel, gut, intestine, and even human-on-a-chip that integrates multiple organs have been engineered as microfluidic devices for pharmaceutical research. Relevant studies have been systematically and comprehensively reviewed elsewhere in recent years [153,154].

However, the conventional construction of OOC usually relies on the precise processing and manufacture of polydimethylsiloxane (PDMS) (e.g., STL, hot embossing, laser direct-writing, and micro-molding), followed by selective cell-seeding, which remains complicated, time-consuming, and expensive. However, PDMS can only be used as an element for building the reaction device rather than simulating ECM for encapsulating cells; therefore, it is difficult to establish a 3D heterogeneous construct. Although PDMS is a nontoxic and biochemically inert material, its active absorbance of small molecules (less than 500 Da) can largely affect the test results of relevant drugs [155]. In contrast, 3D bioprinting can directly fabricate living constructs with embedded perfusable microchannels using a wide range of polymeric biomaterials and multiple cell types to produce heterogeneous tissue/organ mimics as complex OOCs. 3D bioprinting is considered a practical approach, given advantages such as convenience and flexibility, and has been employed for manufacturing diverse OOC platforms for evaluating drug safety and efficacy.

#### 4.3.2. Strategies for 3D Bioprinting of Organ-On-A-Chip

One essential goal of OOC 3D bioprinting is the incorporation of microfluidic channels with patterned living constructs to dynamically provide metabolic supplies and biochemical/mechanical stimulations simulating natural microenvironments in the human body. Pioneering investigations have explored various fabrication strategies for building microchannels, including (1) direct fabrication of fluidic chambers, (2) micro-molding, and (3) coaxial printing of hollow tubes.

Extrusion- and lithography-based bioprinting were applied to directly create microchannels using printable polymeric materials. Lee et al. reported a one-step construction process that deposits multiple types of living cells within a prefabricated chamber by 3D extrusion of poly(ε-caprolactone) (PCL) (Figure 5A) [91]. This approach not only eliminated the secondary cell-seeding procedure commonly needed in conventional OOC construction but significantly reduced protein absorption when PCL material was applied, leading to accurate measurement of metabolism and drug sensitivity. However, the limited resolution of the extrusion-based 3D bioprinting technique (>200 μm) restricts the dimensional minimization of the fabricated device; thus, it is difficult to create refined capillary-like channels that mimic physiological conditions in natural tissues and organs. Lithography is another strategy that can be used to build microfluidic devices immediately. Owing to the high resolution of the laser source, the resolution of the printed structure is markedly superior to that produced by the micro-extrusion-based 3D bioprinting technique. For instance, using the STL technique, Chen et al. fabricated various microfluidic channels such as arborescent and hierarchical vascular networks of varying diameters, ranging from thousands to dozens of micrometers (Figure 5B) [156]. Despite the capacity of lithography for building microchannels, most photocurable bioinks rely on UV irradiation for crosslinking, which might be detrimental to cell viability due to the damage to cellular DNA. Hence, a common approach involves seeding endothelial cells onto the luminal side of the channels to mimic vascular function, which leads to difficulties in engineering heterogeneous constructs with multiple cells. Optimizing the UV crosslinking process and selecting visible-light-curable materials could be a viable solution to this problem.

3D bioprinting of fugitive templates with sacrificial materials (e.g., gelatin, agarose, Pluronic F127 (PF127)) is an effective approach for defining channels with complicated patterns and networks within hydrogel constructs. Upon seeding endothelial cells, microchannels that exhibit vascular functions can be acquired. Based on this methodology, Wu et al. developed perfusable vascular networks of 200–600 μm in diameter by printing PF127 within a PF127 diacrylate (PF127-DA) reservoir [157]. By employing UV irradiation to crosslink the PF127-DA, the fugitive PF127 was liquefied and evacuated, resulting in a built-in channel hydrogel. In addition to single channels, complicated and correlated tubular tissue mimics can be realized by relying on the flexibility of the 3D bioprinting technique. Lin et al. created 3D vascularized proximal tubule models embedded in a permeable ECM hydrogel composed of adjacent conduits lined with epithelial and endothelial tissues (Figure 5C) [158]. This kind of OOC allows the coculture of renal and vascular tissues to investigate epithelial and endothelial tissue crosstalk, such as the reabsorption process in tubular-vascular exchange, responses to hyperglycemic conditions, and endothelial dysfunction. Despite the attractive capacity for engineering microchannels, the indirect fabrication process usually requires cell-seeding as an additional procedure to achieve vascular function, which could result in low efficacy or deficient endothelium formation.

As another strategy, coaxial bioprinting techniques can directly produce perfusable tubes by co-extruding bioinks and crosslinkers through a core/shell nozzle. In this method, alginate-based bioinks are commonly used owing to their instant crosslinking behavior when encountering Ca^2+^ ions [159]. However, because alginate lacks cell-adhesive moieties, bioactive materials are usually supplemented to improve their cell-friendly performance. Accordingly, Gao et al. suggested a hybrid bioink composed of vascular tissue-specific bioink and alginate (Figure 5D) [93]. By coaxially extruding it with PF127 and Ca^2+^, vasculature-on-a-chip capable of recapitulating the physiological functions and inflammatory responses of endothelial tissues was successfully 3D bioprinted [160]. Other materials such as GelMA can also be used to manufacture cell-laden microtubes [161]. In addition to relying on rapid ionic gelation, in situ photo-crosslinking is another approach to realize the direct biofabrication of microtubes [162]. Nonetheless, coaxial bioprinting is generally ‘a one-stroke drawing’ method that cannot create channel networks with interconnected joints. Only a few studies have reported the possibility of coaxial printing of bifurcated vasculature by controlling the gelation time of the bioink. Hence, further efforts are necessary to broaden the application of coaxial bioprinted tubes to construct complicated OOCs.

#### 4.3.3. Pharmaceutic Applications of 3D Bioprinted Organ-On-A-Chip

Owing to the capacity of 3D bioprinting for automated programming of sophisticated tissue architectures, organized cell/ECM orchestrations, and microenvironmental stimulations, personalized OOC has emerged.

Typically, generalized therapeutic strategies are employed for disease treatment. However, establishing individualized platforms to explore optimal therapy for specific medical cases is as important as creating standardized tools to understand general information regarding diseases. Considering cancer as an example, many cancer-on-a-chips have been developed to examine tumorigenesis, tumor angiogenesis, tumor metastasis, and tumor interactions [163]. However, in this patient-specific disorder, conventional “one-size-fits-all” treatments usually demonstrate varying degrees of therapeutic efficacy and side effects, which might lead to treatment failure. Hence, patient-specific customized modeling is an advisable choice for precision medicine. Yi et al. 3D bioprinted glioblastoma-on-a-chip using endothelial cells and patient-specific glioblastoma cells, encapsulated in a decellularized porcine brain bioink that emulates the ECM environment in the brain [22]. The chip design helped create three distinct oxygen gradient regions with compartmentalized cancer-stroma structures that mimic glioblastoma pathological conditions. This model was found to be viable for identifying potential drug combinations for patient-specific treatment.

Given the limitations in fabrication efficiency and resolution, 3D bioprinting remains incapable of producing in vitro diagnostic or screening platforms for HTS (10,000–100,000 compounds test per day) [164]. In an attempt to overcome this bottleneck, Hwang et al. proposed an integrated 3D bioprinter based on microscale continuous optical printing [165]. The developed printer allowed the fabrication of structures with varying spatial geometries and tunable mechanical properties with high reproducibility [166]. Accordingly, it is possible to produce massive living biological samples within multiple-well plates in a short manufacturing time (20–40 min to fill a 96-well plate with the 3D bioprinted constructs completely). Although this method has not been verified for constructing multifluid platforms, the rapid generation of in vitro 3D tissue models within industrial multiple-well cell culture plates would accelerate the achievement of 3D bioprinted high-throughput OOC.

## 5. Drug Delivery System

Drug delivery systems are engineered formulations for targeted delivery and/or controlled release of therapeutic agents. Drug delivery through oral/rectal/vaginal routes are representative routes for drug administration into the circulatory system; however, they affect the entire body through the cardiovascular, respiratory, gastrointestinal, and nervous systems. Transdermal and surgical drug delivery systems are strategies that directly deliver drugs to the site of action for release and absorption. Such approaches aim to improve the bioavailability of drugs at disease sites. We discuss each drug delivery system in the following subsections.

### 5.1. 3D Printed Drug Release System

#### 5.1.1. Oral/Rectal/Vaginal Drug Delivery System

Oral/rectal/vaginal-based drug administration primarily takes place via absorption of the drug in the gastrointestinal tract, and a variety of formulations have been developed, including tablets, gels, capsules, lozenges, pills, and pellets. In particular, oral-based drug delivery is the most well-established and a preferred route for drug administration for treating numerous diseases, owing to its ease of application and superior patient compliance [167]. Orally delivered drugs have highly variable mechanisms of action, as drugs are metabolized by the liver or eliminated by the kidney during delivery into the bloodstream. The aqueous solubility of the drug compound in the gastrointestinal tract is an important factor determining the bioavailability of orally delivered drugs. Therefore, the physiological environment of the digestive system should be considered when designing a drug formulation to effectively deliver oral-based drug carriers. Conversely, rectal- or vaginal-based drug delivery affords immediate effects with a faster onset of action when compared with the oral route, as these routes escape primary metabolism in the gastrointestinal system [168].

These drug delivery routes commonly involve sustained drug release to enhance drug solubility or reduce the incidence of side effects. However, altered drug absorption due to variable drug permeability and interactions with gastrointestinal contents, as well as metabolic processes, can result in poor bioavailability and significant fluctuations in plasma drug levels [169]. These fluctuating drug levels often result in an increased incidence of adverse effects by exceeding the appropriate plasma levels or decreased drug efficacy due to sub-therapeutic levels [169,170]. These limitations in traditional drug delivery systems can be surpassed by manufacturing personalized drugs considering the mechanism of absorption of the administration route and the medical status of each patient. Traditional manufacturing processes for personalized medicine have raised concerns regarding cost-effectiveness due to substantial inaccuracies and time consumption, as they inevitably involve manual procedures. To overcome conventional limitations, the introduction of 3D printing enables the creation of customized drug delivery devices while accounting for various individual characteristics.

Taken together, 3D printing technology has been receiving considerable attention as a promising strategy to replace the traditional mass-production-based methodology by designing a personalized drug with diverse printing materials and 3D shapes according to delivery routes and absorption mechanisms [171].

#### 5.1.2. Strategies for 3D Printing of Oral/Rectal/Vaginal Drug Delivery System

The transition to a more personalized treatment strategy that considers the medical condition and characteristics of each patient is currently an important issue in medicine and pharmaceutical development. Accordingly, the introduction of 3D printing is of utmost importance for formulating predictable and reliable patient-centered drug delivery devices with tailored sizes, shapes, and release profiles. In particular, determining the onset and release rate of a drug are crucial factors in developing drug delivery systems. Recent studies have explored strategies for establishing advanced delivery devices by (1) utilizing polymers with various drug release profiles and (2) designing drug carriers based on computer simulation.

The use of biodegradable polymers that enable spatiotemporal control of drug release plays a vital role in reducing drug toxicity and improving the therapeutic efficacy of drug delivery systems [171]. The development of printable synthetic polymers with diverse drug release mechanisms has substantially advanced drug delivery research. Lee et al. and Yoo et al. illustrated that matrices constructed of polysaccharides, poly(amino acids), polyesters, and polyamides release drugs based on hydrolysis or enzymatic breakdown of amide, ester, and hydrazone bonds in their backbones [172,173]. Drug delivery systems comprising polyorthoesters and polyanhydrides typically cause surface decomposition, as surface polymer erosion is faster than water diffusion delivered into the system core [174,175]. In contrast, drug carriers, consisting of PCL, poly(lactic acid) (PLA), and poly(lactic-co-glycolic acid), are known to undergo bulk degradation at the core and surface of carriers, resulting in a relatively fast degradation rate.

It has been reported that computer-aided drug design facilitates rapid exploration, optimization, and implementation of drug development processes through simulation-based analysis, dramatically reducing time and costs [176]. The computer-based approach utilizes a software program to establish standards that associate biological drug activity with the compound structure [177]. Imam et al. illustrated the capability to design new drug molecules that exhibit improved interactions with target proteins by computationally simulating bio-affinity during the structure-based drug design [178]. Skowyra et al. employed commercially available computer software, Autodesk^®^ 3ds Max^®^ Design 2012, to analyze the disintegration kinetics of a poly(vinyl alcohol) (PVA)-based drug carrier [179]. The data analyzed determined the dosing accuracy and the geometry of the drug system, achieving significant improvement. The authors confirmed that the precision of dose control ranged between 88.7% and 107%, affording uniform drug release during 24 h of administration when the drug delivery system was generated by a digitally controlled 3D printer.

#### 5.1.3. Pharmaceutic Applications of 3D Printed Oral/Rectal/Vaginal Drug Delivery

Exploratory research on 3D printing in pharmaceutics has revealed the immense potential of personalized formulations for each drug delivery route (Table 2). Lim et al. printed a carbamazepine-loaded oral drug device for tonic seizures that maintained a constant effective surface area using an acrylonitrile butadiene styrene filament, known to be stably biodegradable (Figure 6A) [180]. The device achieved zero-order drug release kinetics of antiepileptic drugs, improved patient compliance, achieved effective therapeutic doses, and minimized side effects. In addition, Hsu et al. confirmed that the 3D printing technology could produce highly reliable solid dispersions with precise dosage control of active pharmaceutical ingredients [181]. The study used PEG with an excellent biodegradation profile and naproxen, an anti-inflammatory agent, as a model drug and successfully developed a drug delivery system with uniform composition and sustained-release kinetics.

Immediate drug release is a particularly important factor to consider when formulating a drug that requires a rapid effect, such as pain relief. From this perspective, Okwuosa et al. developed a caplet-shaped oral drug system using extrusion-based printing technology for immediate drug release [182]. Hydrophilic pharmaceutical-grade polymers such as poly(vinyl pyrrolidone) (PVP) or Eudragit^®^ EPO have been commonly utilized for immediate drug release to create 3D printed drug devices, as they rapidly dissolve in contact with body fluids. In order to deliver the drug into the intestine, it is necessary to prepare a system protected from the potently acidic stomach environment while dissolving in an alkaline environment. Goyanes et al. confirmed that PVA-based capsules containing budesonide coated with an enteric polymer, Eudragit^®^ L, can be effectively delivered through oral route to the small intestine to treat inflammatory bowel disease (Figure 6B) [183]. Okwuosa et al. engineered gastric-resistant devices loaded with theophylline, budesonide, and diclofenac sodium using a dual 3D printing extrusion method [182]. The developed oral devices consisted of a methacrylic acid copolymer and PVP at the shell and core, respectively. This shell-core design system demonstrated sufficient resistance at high acid levels and initiated drug release inside the core in an alkaline environment, releasing up to 80% of the loaded drug within 8 h.

The ability to readily design complex geometries using 3D printing facilitates the fabrication of drug delivery devices with desired release profiles and multiple features. A recent study has demonstrated that a structure in the torus form can stably and sustainably release drugs by maintaining a constant surface area-to-volume ratio. Wang et al. successfully printed drug devices loaded with paracetamol and 4-aminosalicylic acid (4-ASA) in torus form via STL (Figure 6C) [184]. The dynamic dissolution simulation of the gastrointestinal tract revealed sustained drug release independent of pH, suggesting a new 3D printing platform for drug development. In a similar attempt by Yu et al., a multi-layered drug system was developed to provide a linear release profile based on a computer-aided design (CAD) model. As a model drug, a hydroxypropyl methylcellulose-based acetaminophen-loaded system was found to modify the release profile based on structural dimensions through in vitro dissolution analysis (Figure 6D) [185]. Sun et al. formulated analgesic suppositories of various shapes and sizes using non-soluble silicone elastomers by employing a stereolithographic approach (Figure 6E) [186]. The suppository molds for rectal and vaginal routes were first created using a commercial 3D printer based on CAD software. Drugs and silicone polymers were then injected into the polypropylene molds. The authors observed that the mechanical properties of the drug-containing silicone elastomer and the rate of drug release could be tuned through the precise formulation of the drug-polymer. In addition, Tudela et al. engineered a cervical cerclage pessary for cervical incompetence and prevention of preterm birth using a similar method [187]. Cervical length and radius were measured with prenatal ultrasound and subsequently converted to the STL format to print the personalized pessary. An attempt was made to develop hormone-eluting devices based on patient-specific anatomy using biodegradable materials. According to Tappa et al., the PCL vaginal suppository containing estrogen and/or progesterone could be tailored to the specific needs of the patient in terms of hormone dosage and duration of hormone therapy [187]. Collectively, the recent advances in 3D printing methodologies offer potential solutions to create oral/vaginal/rectal drug delivery systems capable of customized drug release with desired degradation profiles and geometrics, thereby improving efficacy and increasing patient compliance. These advantages of 3D printing have enabled the commercialization of the first FDA-approved 3D printed drug, Spritam, in 2015 and ZipDose^®^ by Aprecia^®^ Pharmaceuticals [188].

**Table 2 pharmaceutics-13-01373-t002:** Representative pharmaceutical application of 3D printed oral/vaginal/rectal drug delivery.

	3D Printing Strategy	Achievement	Reference
Oral drug delivery system	Piezoelectric inkjet printing with drug dissolved in propylene glycol	Automated engineering of medicines; precise patterning of porous substrates and dosing of low-dose drug substances	[189]
	Piezoelectric inkjet printing with drug mixed in pre-polymer solution	Immediate drug release profile owing to hydrophilic active materials for accurate dosing	[190]
	Extrusion-based printing with drug loaded filament	Coated with a layer of enteric polymer to prevent drug degradation in acidic pH	[183]
	Dual extrusion-based 3D printing	Core-shell designs for delayed-release kinetics; drug with high water solubility and gastric-resistant products	[37]
	Extrusion-based 3D printing	Five-in-one dose combination polypill with controlled release	[46]
	STL with drug mixed in pre-polymer	Torus-shaped drug carrier for customized drug release profile	[184]
Vaginal/rectal drug delivery system	Extrusion-based 3D printing with estrogen and/or progesterone	Personalized implants and devices for obstetric and gynecological applications.	[191]
	Extrusion-based 3D printing with drug mixed in pre-polymer	Custom-made T-shaped intrauterine systems and subcutaneous rods	[192]
	STL with drugs mixed in pre-polymer	Non-dissolving suppository/pessary with tunable and sustained drug release	[186]

### 5.2. 3D Printed Transdermal/Surgical Drug Delivery System

#### 5.2.1. Transdermal/Surgical Drug Delivery System

Transdermal drug delivery is a method for delivering medications by applying a formulation to intact skin. This route affords the advantage of bypassing metabolic processes, reducing the burden of pill administration, and improving patient compliance [193]. In particular, transdermal drug delivery has been employed as a typical route of vaccination, as a plethora of dendritic cells is available in the skin [194]. In addition to the transdermal route, implantable drug delivery devices, including surgical patches, disks, and stents, to deliver drug compounds enable targeted drug delivery, resulting in excellent therapeutic outcomes even at relatively low drug concentrations [195,196]. These features minimize the potential off-target side effects and improve patient compliance; therefore, surgical application with an implantable device is recognized as a promising treatment tool [197].

Surgical implants require a customized design that satisfies various considerations, such as anatomical differences according to patients, age, sex, and disease state. In this context, commercially available surgical devices generally lack a technological approach for creating drug-eluting implants that outline the patient’s anatomy. However, recent advances in 3D printing have allowed the fabrication of various types of patient-specific implantable devices containing drugs, providing opportunities for customized dosing in a narrow therapeutic window.

#### 5.2.2. Strategies of 3D Printed Transdermal/Surgical Drug Delivery System

Traditional drug delivery systems have relied on the mass production of average-sized therapeutic devices for a typical patient through implantation or attachment to the lesion. This mass production-based system remains a challenge for implementing various treatment regimens due to individual differences, resulting in low drug efficacy and several side effects in some patients. The introduction of flexible technology that allows the creation of customized devices that reflect the anatomical, pathological, and physical characteristics of a patient is of great interest in the pharmaceutical development of drug system.

A recent strategy using 3D printing technology to manufacture patient-specific pharmaceuticals established complex freeform shapes with a high resolution based on clinical information obtained from CT or MRI scans of the patient. The printing of implants reflecting the patient’s anatomical characteristics through CAD further enhances the competitiveness of narrow therapeutic window drugs. Muwaffak et al. successfully printed a patch-shaped structure that accurately reflected the patient’s nose contour through STL technology based on facial data obtained from a 3D scanner [198]. Another 3D printing strategy for surgical drug delivery developed implants using biodegradable materials safely degraded and naturally released in the human body. A stent, for example, is a tube-shaped surgical device inserted into a blocked passageway of the artery, trachea, or esophagus to keep it open, restoring the flow of blood or other fluids. Although the superior mechanical properties of traditional metal-based permanent stents prevent stenosis of organs subjected to harsh physical interference, extraction is required to avoid infection and tissue damage total drug elution. As successive invasive surgery is considered a significant burden to patients, the current study of 3D printed stents mainly focused on the use of biomaterials that can safely disintegrate within the body. Ha et al. developed a PCL-based esophageal biodegradable stent loaded with an esophagus-derived ECM hydrogel for therapeutic purposes [87]. The authors demonstrated the physical stability of the polymeric stent and the therapeutic effect of the hydrogel without any cytotoxic or adverse outcomes during degradation of the PCL substructure in an esophagitis animal model.

In conclusion, the current strategy in 3D printing utilizing CAD-based biomedical information and printable biodegradable materials offers a straightforward and less expensive solution to create customized devices for transdermal and surgical drug delivery.

#### 5.2.3. Pharmaceutic Applications of 3D Printed Transdermal/Surgical Drug Delivery

Multiple attempts have been made to introduce 3D printing into pharmaceutical studies for establishing topically applicable drug delivery systems (Table 3). The skin is an anatomically undulating structure located on the outermost part of the human body. A fabrication strategy that can adjust the shape and dosage according to the characteristics of individual patients is required, rather than a mass-produced system. Goyanes et al. converted the contours of the patient’s face obtained through a 3D scanner into a CAD model to create a patient-specific nose patch for acne treatment using two different printing methods (Figure 7A) [199]. In the extrusion method, a biodegradable synthetic polymer was melted at a high temperature and printed as a nose-shaped mask loaded with salicylic acid. In another method, salicylic acid was dissolved in a photocrosslinkable polymer to create a nasal patch through the STL approach. This patch showed higher resolution and drug loading capacity with no drug degradation when compared with the nasal patch fabricated by the extrusion method.

Microneedle transdermal patches are microscopic applicators that create microscopic pores in the skin to allow drugs such as vaccines and macromolecules to be topically administered and offer an attractive alternative to oral non-steroidal anti-inflammatory drugs. The needle length is approximately 25–2500 µm, which can reach the depth of the skin capillary, and the diameter is designed to prevent nerve damage when attached [200]. These microneedle-based drug delivery systems are of considerable interest as they do not cause pain during self-application. In particular, recent studies have applied 3D printing assisted with a CAD program to precisely recapitulate the curved, undulating skin surface. Based on the CAD image, Lim et al. introduced a dual functional skin splint using a light-assisted 3D printer [201]. The developed transdermal splint administered anti-inflammatory drugs through a microneedle array and served as a splint for physically fixing the finger joints or tendons. As the coating process for drug deposition on the microneedle requires a high degree of precise control, it is substantially difficult to implement using traditional techniques [201]. However, advances in inkjet printing have enabled the accurate deposition of a specific amount of solubilized drug onto needles while maintaining the mechanical strength of the conventional microneedle substrate. Boehm et al., for example, demonstrated the utilization of a piezoelectric-based inkjet printer to coat quantum dots as a drug carrier on the microneedle surface, followed by confirmation of the effective drug delivery in porcine skin [202].

Surgical patches are one of the most widely employed medical tools, as they are relatively simple in terms of fabrication and can provide direct treatment to internal organs. As illustrated by Yi et al., 3D printed biodegradable patches consisting of an anticancer drug, 5-fluorouracil, demonstrated dramatically decreased tumor size in a mouse tumor model with few side effects (Figure 7B) [203]. The developed patch with flexible properties was attached to the tumor site, delivering controlled drug release. In addition, Boetker et al. developed an antimicrobial implant by establishing a 3D printing condition in which thermal degradation does not occur when printing with a mixture of an antimicrobial drug, nitrofurantoin, and PLA at high temperature [204].

Advances in 3D printing technology for transdermal and surgical drug delivery systems have accelerated the development of innovative personalized drug delivery systems that accommodate various patient cases.

**Figure 7 pharmaceutics-13-01373-f007:**
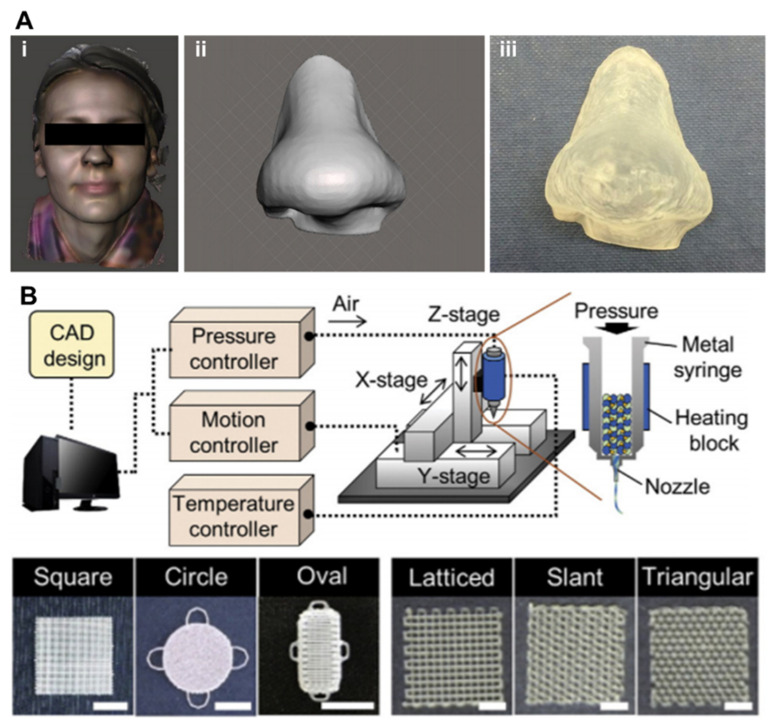
Representative examples of 3D printed transdermal/surgical drug delivery systems. (**A**) Images of the fabrication process for a patient-specific nose transdermal patch. (i) 3D scanning of a volunteer, (ii) 3D model of the nose, and (iii) 3D printing of nose-shaped drug device encapsulating salicylic acid. Reproduced with permission from [199], Elsevier, 2016. (**B**) 3D printing of 5-fluorouracil (5-FU)-loaded surgical patch for pancreatic cancer suppression. Schematic illustration showing in-built extrusion-based 3D printing system (upper). Different shapes of patches with various porous patterns (lower) (scale: 5 mm). Reproduced with permission from [203], Elsevier, 2016.

**Table 3 pharmaceutics-13-01373-t003:** Representative pharmaceutical application of 3D printed localized drug delivery.

	Printing Strategy	Characteristics	Reference
Surgical patch	Piezoelectric inkjet printing with drug mixed in pre-polymer	Tailored dosage forms in a single step with minimal excipients and operations for antibiotics; sustained drug release for 5 days	[205]
	Extrusion-based 3D printing with drug mixed in pre-polymer	3D printing of custom-made and drug-loaded feedstock products for antibacterial medication	[206]
	Extrusion-based 3D printing with drug encapsulated in hydrogel	3D printed xenografts for cancer growth suppression for suppressing pancreatic cancer	[203]
Transdermal patch	Extrusion-based 3D printing with drug loaded filament	Customized design by combining 3D scanning and 3D printing for antimicrobial wound dressing	[198]
	Extrusion-based 3D printing with drug loaded filament	Flexible personalized-shape drug-loaded device by combining 3D scanning and 3D printing for acne on the nose	[199]
	STL-based 3D printing	Two-photon polymerization, microfabrication, and subsequent PDMS micro-molding process	[207]
	STL-based 3D printing	Personalized curved surfaces for drug delivery and splinting of finger	[201]
Surgical stent	Extrusion-based 3D printing with drug mixed in pre-polymer	Direct 3D printing of biodegradable polymer–graphene Composite with dual drug incorporation Direct 3D printing of biodegradable polymer–graphene Composite with dual drug incorporation Direct 3D printing of biodegradable polymer–graphene Composite with dual drug incorporation Direct 3D printing of biodegradable vascular stent with dual drug incorporation	[208]
	Extrusion-based 3D printing with drug encapsulated in hydrogel	Engineering of an esophageal stent using esophageal-specific bioink to provide tissue-specific microenvironments for therapeutic effects	[87]
	STL-based 3D printing	Bioresorbable and drug-eluting vascular stent	[209]

## 6. Conclusions

3D printing is broadly defined as the process via which a 3D model is used to generate a physical object, and it has recently garnered considerable attention due to growing applications. In recent years, the development of numerous biomaterials has resulted in advances in 3D printing, thus, making it a novel engineering tool that can build 3D objects to be applied in the pharmaceutical industry. Furthermore, 3D bioprinting has been widely introduced as a method of constructing living tissues or organs by employing biological components such as cells and biomolecules.

In traditional pharmaceutical approaches for drug screening, 2D-based cell cultures and animal models have been used to elucidate biological mechanisms of test drugs and uncover disease states. However, these screening models have often failed to comprehensively represent the actual human body due to different microenvironments and genetic discrepancies. The ability of 3D bioprinting to recapitulate living tissues that are substantially similar to natural structures can overcome the limitations of existing preclinical models. Although current pioneering drug screening research has proven the obvious benefits of 3D bioprinting technology, research into patient-specific drug development is still in its infancy. Studies on various characteristics that may appear when using patient cells should be conducted more actively.

Furthermore, the manufacture of drug formulations utilizing 3D printing is expected to play a vital role in the development and innovation of various drug delivery systems. Unlike traditional pharmaceuticals that rely on mass manufacturing, 3D printing approaches can potentially provide individualized treatments to patients, improving efficacy and side effects for individual patients. The study of the 3D printed drug delivery systems currently under investigation are focused on the design depending on the drug release behavior and drug delivery mechanism; thus, 3D printing technology is an effective approach to the formulation of patient-specific drug delivery systems.

Taken together, as a growing number of patients demand customized therapeutics, 3D printing technology is well suited to offer tailored drug screening platforms and drug delivery systems that fulfill a set of individualized needs. Given its potential applications in several medical domains, 3D printing is uniquely suited to assist biologists and pharmacologists at different stages of drug development and patient treatment.

## Figures and Tables

**Figure 1 pharmaceutics-13-01373-f001:**
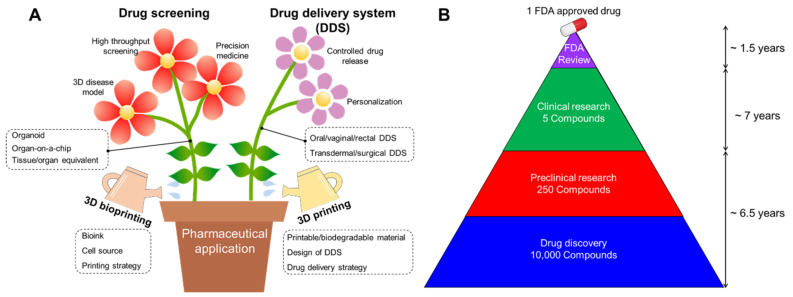
(**A**) Schematic illustration showing advances in pharmaceutical applications, including drug screening and drug delivery system (DDS) by employing 3D printing and bioprinting technology. (**B**) Average drug development timeline for one FDA-approved drug.

**Figure 2 pharmaceutics-13-01373-f002:**
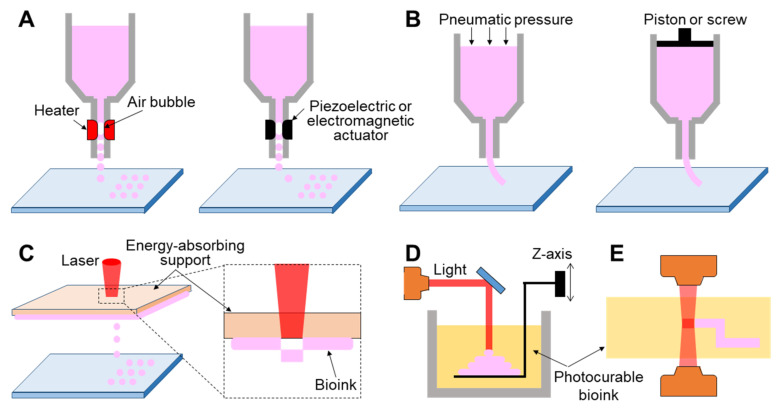
Schematic illustration showing printing strategy. (**A**) Inkjet-based 3D printing, (**B**) extrusion-based 3D printing, and light-assisted 3D printing, including (**C**) laser-induced forward transfer technique, (**D**) stereolithography, and (**E**) photon polymerization technique.

**Figure 3 pharmaceutics-13-01373-f003:**
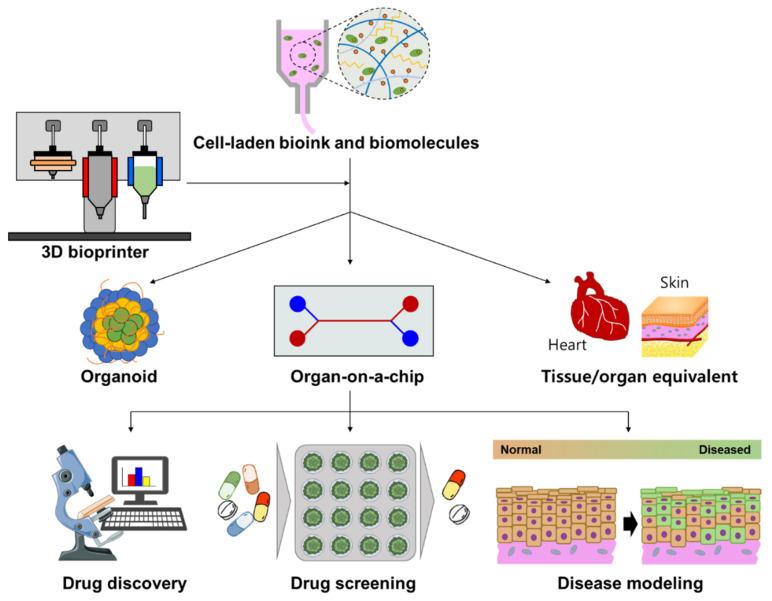
Schematic illustration representing 3D bioprinting, enabling the establishment of 3D cell culture devices, including organoid, organ-on-a-chip, and tissue/organ equivalent, that can advance pharmaceutical applications of drug discovery, drug screening, and disease modeling. This figure was prepared using a template on the Sevier medical art website (http://www.sevier.fr/sevier-medical-art, accessed on 12 July 2021).

**Figure 4 pharmaceutics-13-01373-f004:**
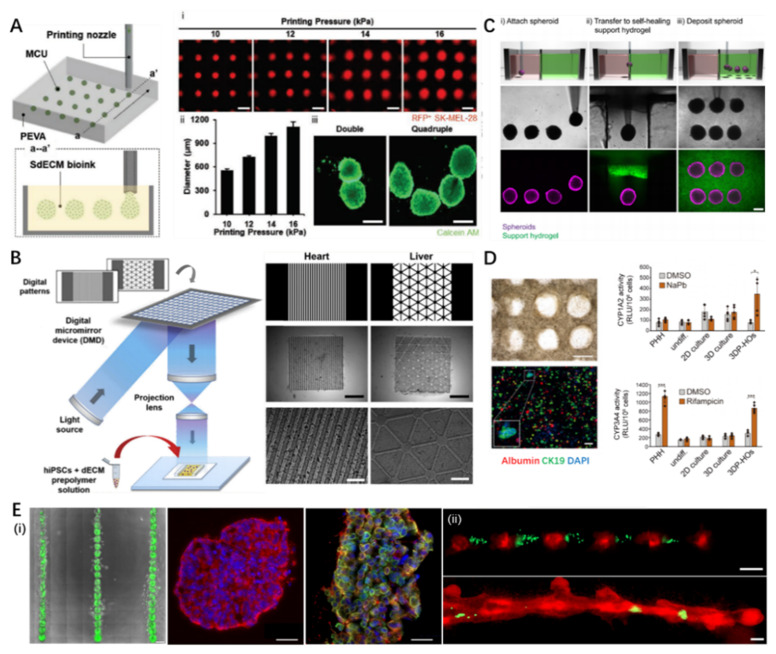
Representative 3D bioprinted organoids and pharmaceutic applications. (**A**) The concept of in situ 3D bioprinting of cell aggregates and its potential for in situ printing of cancer organoids: (i) size control of spheroids by altering printing pressures; (ii) quantification of spheroids dimensions varied with printing pressure; (iii) stained images of double/quadruple melanoma organoids (scale: 500 μm). Reproduced with permission from [116], John Wiley and Sons, 2021. (**B**) Schematic and image demonstrating the 3D bioprinted spheroids in self-healing support hydrogels. Reproduced with permission from [117], Elsevier, 2019. (**C**) STL-based rapid 3D bioprinting process to fabricate dECM tissue constructs with tissue-specific iPSCs and biomimetically patterned heart and liver dECM tissue constructs. Reproduced from [121], Springer Nature, 2021. (**D**) 3D bioprinted hepatorganoids on day 10 and the P450-Glo assay of cytochrome activity (CYP1A2, CYP3A4) in the hepatorganoids after 7 days of differentiation. Reproduced with permission from [123], BMJ Publishing Group Ltd., 2021. (**E**) (i) 3D bioprinted patterned tumoroids and desired tumor cytokine expression (red: CK-8, green: CK-5, scale: 100 μm), (ii) alternative 3D bioprinting of MDA-MB-468 cells (green) and MCF-12A cells (red) at day 1 (up) and day 7 (down) demonstrating incorporation of cancer cells into the organoid (scale: 200 μm). Reproduced from [130], Scientific reports, 2019. STL, stereolithography; dECM, decellularized extracellular matrix; iPSCs, induced pluripotent stem cells.

**Figure 5 pharmaceutics-13-01373-f005:**
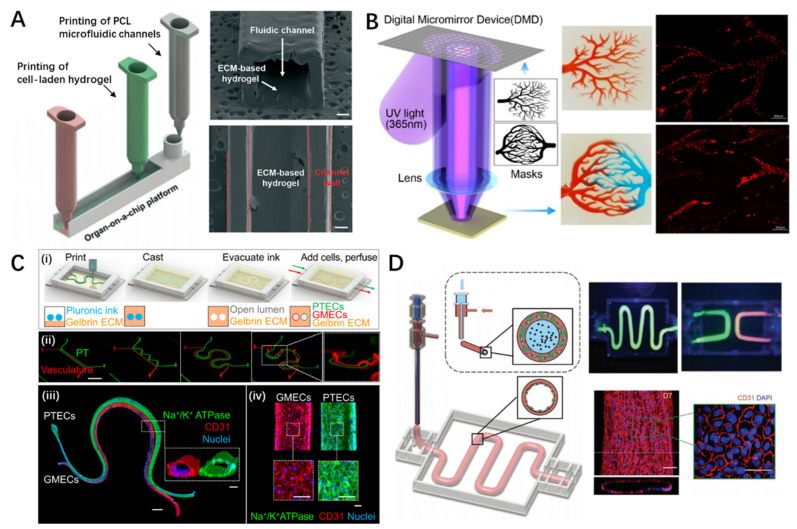
Representative 3D bioprinted organ-on-a-chip and pharmaceutic applications. (**A**) Schematic and optical images of directly 3D printed microfluidic chips (scale: 350 μm). Reproduced from [91], The Royal Society of Chemistry, 2016. (**B**) Schematic of the projection-based 3D printing platform and images showing the capillary action of channels and cell distribution in the capillary-like scaffold. Reproduced with permission from [156], American Chemical Society, 2018. (**C**) (i) Schematic of 3D vascularized proximal tubule model fabrication process, (ii) design and 3D bioprinted simple/complex models, (iii) whole-mount immunofluorescence staining of the 3D tissue (scale: 1 mm and 100 μm for inset), and (iv) high-magnification images of modeled tissues after staining (scale: 100 μm). Reproduced from [158], Proceeding of the National Academy of Sciences, 2019. (**D**) Schematic diagram of the construction of freestanding functional vascular models with complex patterns and the maturation of printed vasculatures (scale: 100 μm). Reproduced with permission from [93], John Wiley and Sons, 2018.

**Figure 6 pharmaceutics-13-01373-f006:**
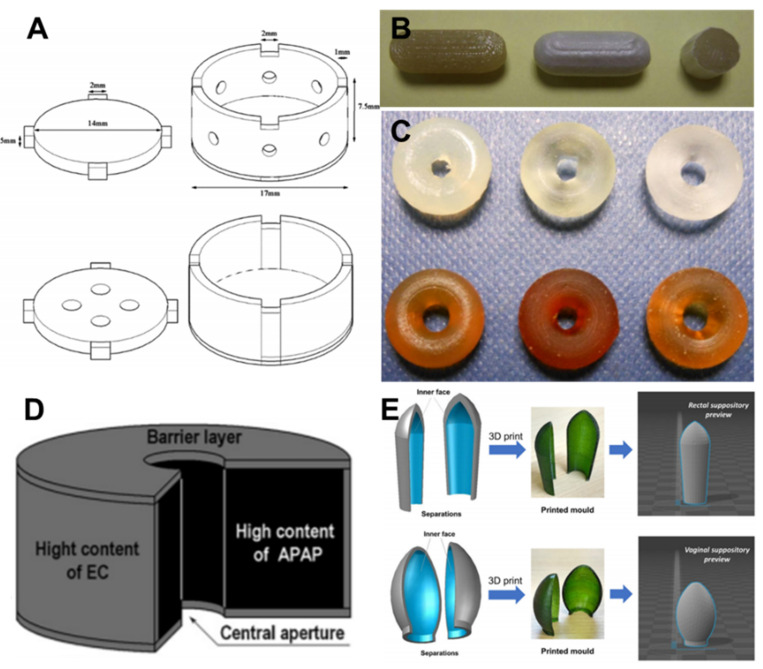
Representative examples of 3D printed oral/vaginal/rectal drug delivery systems. (**A**) 3D model of the carbamazepine-loaded oral drug device for tonic seizures. Reproduced with permission from [180], Elsevier, 2016. Photograph of 3D printed oral drug carriers loaded with (**B**) budesonide (reproduced with permission from [183], Elsevier, 2015), (**C**) (upper) paracetamol, and (lower) 4-aminosalicylic acid (4-ASA). Reproduced with permission from [184], Elsevier, 2016. (**D**) Schematic illustration of torus-shaped multi-layered drug delivery device. Reproduced with permission from [185], Elsevier, 2009. (**E**) Computer-aided designs of suppository molds for (upper) rectal and (lower) vaginal and rectal routes. Reproduced with permission from [186], Elsevier, 2016.

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
