# Peer review of "3D Printing of Pharmaceutical Application: Drug Screening and Drug Delivery"

_pharmaceutics, 2021, doi:10.3390/pharmaceutics13091373_

Round 1
Reviewer 1 Report
In the present MS, an extensive overview of pharmaceutical applications employing 3D bioprinting technology, focusing on drug screening and drug delivery systems, was reviewed. However, the relationship of drug screening and drug delivery was unclear, is it coordinative relationship or progressive relationship.
Reviewer 2 Report
The present manuscript is a systematic and comprehensive review about pharmaceutical applications of 3D bioprinting, mainly focused on drug screening and drug delivery systems. The review is well written and documented and representative examples of strategies and pharmaceutical applications are reported for each type of 3D bioprinted system.
Specific comments:
- Please review pages in Contents list. Section 5.2 should say 3d bioprinted localized drug delivery.
- Page 4 (line 140): It should say ...we describe...
- Page 9 (line 339): It should say ...range of features...
- Page 9 (line 348): It should say ...depends on various factors...
- Page 16 (line 623): It should say ...and brain...
- Page 21 (line 764): It should say ...two opposed microchannels...
- General information in Section 5.1.1. Systemic Drug Delivery could be summarized.
Reviewer 3 Report
My general comment is that the manuscript "3D Bioprinting of Pharmaceutical Application: Drug Screening and Drug Delivery" is interesting and the topic introduced is currently important.
However there are several important points and concepts that need to be revised .
First of all the authors should clearly define what is 3Dprinting and 3Dbioprinting because they are confusing them and sometimes is ot clear what they are referring to.
Secondly some figures are too much crowded and not clear.
Here below my detailed comments:
Page 3, line 88-90: Please define 3D-bioprinting and explain the difference between 3Dprinting and 3D-bioprinting: 3D-bioprinting involves use of biologic material such as cells. The authors should better explain.
Page 4, lines 133-134: The authors should better explain how 3D-bioprinting is linked to drug delivery systems, taking into account that 3D-bioprinting involves printing of cells. It is quite clear how 3D-bioprinting can be applied to in vitro drug screening, but it is not clear how it applies to drug delivery systems.
Page 5, Figure 1A: In my opinion Figure 1A is not representative and it is missleading. I would eliminate it or substitute it with a more explanatory one.
Page 7, line 248 (Chpt 2.2.1.): The authors should better focus on controlled drug release formulation obtained by 3Dprinting. The 2.2.1 chpt should be completely revised by focusing on 3Dprinted formulations which promote drug controlled release.
Page 8, line 302-303: The authors should better explain how the formulation was manufactured by 3Dprinting.
Page 8, line 310: 3Dbioprinting definition should be introduced above in the paper introduction.
Page 12, lines 498-500: implantable and anatomically patient-specific devices are not 3Dbioprinting applications, but they are 3Dprinting applications. The authors should better clarify.
Page 14, Figure 3: Presently arrows are linked only to OOC, the figure should be completed by indicating which are the outputs of organoids and Tissue/organ equivalent respectively.
Page 24, Figure 5: In my opinion this figure is too much crowded.
I suggest to split it into 3 figures, or to simplify it.
Page 25, line 909: In my opinion the authors should revise the whole chpt 5.1 taking into account that drug delivery systems are 3Dprinted, but they are not 3Dbioprinted. Accordingly the word 3Dbioprinted should be substituted with 3Dprinted along the whole chapter 5.1.
Page 25, line 943: 3DBioprinting is not appropriate here because drug delvery systems are 3Dprinted. In fact the 3Dprinted drug delivery systems do not include cells in their formulation.
Page 26, line 959: Drug systems is not an appropriate definition. The correct definition is drug delvery system. The authors should correct.
Page 26, line 977: As reported above, drug system should be replaced by "drug delivery system".
Page 26, line 980: Please substitute 3Dbioprinting with 3Dprinting
Page 27, lines 1009, 1016: Please substitute 3Dbioprinting with 3Dprinting.
Page 27, line 1039: There are some 3Dprinted drug products on the market. the autors should report this in the chapter.
Page 29, line 1051: As previously underlined, the authors should change the word 3Dbioprinted with 3Dprinted because drug delivery systems do not include cells. The authors should revise all Chpt 5.2 accordingly.
Page 29, line 1051: I suggest to modify "drug delivery" as "drug delivery systems".
Page 27, line 1057: I totally disagree with this definition because transdermal drug delivery aims to systemic administration of a drug through its transdermal absorption. Therefore, transdermal drug delivery cannot be included in localized drug delivery.
Page 31, line 1108: The title should be modified as 3Dprinted localized drug delivery systems"
Page 31, line 1122: As previously underlined, this application do not rely in local drug delivery because it is aimed to systemic effect of drugs.
Page 33, line 1163, conclusions: The conclusions should be rewritten because are a summary of what reported in the paper. The authors should report in the conclusions which is their opinion on the literature revised, which are the industrial outputs of the researches introduced and the future perspectives of 3Dprinting and 3Dbioprinting in the pharmaceutical area.
Page 33, line 1170: I disagree with this definition because 3Dbioprinting involves cell printing.
Round 2
Reviewer 3 Report
The paper has been suitably revised.
The topic is extensively reviewed and updated.